# Structural Optimization and Performance of a Low-Frequency Double-Shell Type-IV Flexural Hydroacoustic Transducer

**DOI:** 10.3390/s24144746

**Published:** 2024-07-22

**Authors:** Jinsong Chen, Chengxin Gong, Guilin Yue, Lilong Zhang, Xiaoli Wang, Zhenhao Huo, Ziyu Dong

**Affiliations:** 1Jiangsu Provincial Institute of Marine Resources Development and Research, Jiangsu Ocean University, Lianyungang 222005, China; 604470816@163.com (C.G.); yueglin@163.com (G.Y.); wangxl@jou.edu.cn (X.W.); hzhlllll99@gmail.com (Z.H.); dongz@jou.edu.cn (Z.D.); 2Department of Mechanical Engineering, Hefei Gongda Vocational and Technical College, Hefei 230016, China; lilong0119@sina.com

**Keywords:** low frequency, double shell, flexural tension transducer, structural optimization

## Abstract

To amplify the displacement of the radiation shell, a double-shell type-IV curved hydroacoustic transducer was proposed. Through Ansys finite element simulation, the vibration modes of the transducer in different stages and the harmonic response characteristics in air and water were studied, and the bandwidth emission of the hydroacoustic transducer was achieved. By optimizing the size of each component, the resonant frequency of the transducer is 740 Hz, the maximum conductivity was 0.66 mS, and the maximum transmitting voltage response was 130 dB. According to the optimized parameters, a longitudinal acoustic transducer prototype was manufactured, and a physical test was conducted in an anechoic pool. The obtained resonant frequency was 750 Hz, the maximum conductivity was 0.44 mS, the maximum transmitting voltage response was 129.25 dB, and the maximum linear dimension was 250 mm, which match the simulated value of the virtual prototype and meet the expected requirements.

## 1. Introduction

Hydroacoustic transducers are an important device used in the analysis of underwater acoustic radiation [1,2,3,4,5]; low-frequency sound waves can be transmitted further underwater [6,7,8,9,10], but the traditional low-frequency underwater acoustic transducer has the disadvantages of structural size [11,12,13,14,15,16], a complex shell [17,18,19,20,21,22], and difficulty in processing [22,23,24,25,26,27]. The nested double-shell structure is thus proposed. Aimed at solving the disadvantages of traditional low-frequency underwater acoustic transducers, such as large structure size and difficult to process complex shell. A type-IV curved underwater acoustic transducer with a double-shell structure is put forward. The resonant frequency of the transducer can be lowered without increasing the structural size of the transducer. Three-dimensional (3-d) printing technology can improve production efficiency; control the cost and reduce the waste of resources; realize large-size, high-precision, multi-field product printing that is unaffected by time and space, improving production efficiency; a complex model can be printed, which improves the freedom of product creation [28,29,30,31,32,33]. One can utilize plastics, metals, ceramics, and other materials as well as a variety of composite materials for mixed printing. These advantages provide a new direction for us to solve the traditional underwater acoustic transducer. Therefore, it is of great significance to introduce the 3-d printing additive manufacturing technology into the production of each component of an underwater acoustic transducer.

## 2. Finite Element Analysis

### 2.1. Finite Element Modelling

The main structure of the double-shell type-IV bending tension transducer includes a piezoelectric ceramic sheet, conductive copper sheet, conductive copper wire, potting resin, inner shell, outer shell, shell cover plate, etc. The 3-d model was obtained through some simplification of the real object, as shown in Figure 1a. Due to its symmetrical structure, to reduce the overall computational burden, a one-eighth-scale physical model (Figure 1b) was used for subsequent calculations.

### 2.2. Modal Analysis

Through the Block Lanczos-mode extraction method, the first four modes of the one-eighth-scale model were extracted (Figure 2). In the first-order vibration mode, the point of maximum displacement of the transducer is in the short axis of the housing and the direction is inward (Figure 2a). The inner and outer shells undergo linear bending vibrations to achieve an optimal radiative effect. At this time, the frequency of vibrations in the transducer is the resonant frequency, and the vibration mode is the ideal working mode. In Figure 2b, the transducer is in the second-order vibration mode, and the maximum displacement of the outer shell occurs at the short end and the direction of this is inward. The point of maximum displacement in the inner shell is at the end of the short axis, and the direction is inward. Therefore, there is a clear reverse problem in the outer shell, and so it is not selected. As shown in Figure 2c,d, when the transducer is in the third and fourth modes of vibration, the reverse problem of the outer shell is more severe than before, so that effective acoustic radiation may not be formed, so the third and fourth modes are not discussed. Through modal analysis, the first-order bending vibration mode is selected.

### 2.3. Harmonic Response Analysis in Water

The harmonic response is analyzed by establishing the water area model; as water is much denser than air, the additional mass produced is greater and it will cause the natural frequency to decrease. In Figure 3, the transmitting voltage response increases with rising frequency, and when the resonant frequency in the water reaches the maximum emission voltage response, it begins to decline at frequencies beyond resonance.

By measuring the admittance *G* and *B* curves of the transducer in water (Figure 4), the resonant frequency and conductance values in water were found to be significantly reduced compared with those in air.

## 3. Optimization of Transducer Structure

### 3.1. Influences of Structural Parameters in Water

#### 3.1.1. Influences of Structural Parameters of the Shell in Water

The influence of each parameter on the maximum transmitting voltage response is shown in Figure 5.

As shown in Figure 5a, as the height *k*_1_ of the pad block of the inner housing increases, the total mass *M* of the transducer also grows, thus enlarging the maximum emission voltage response; as *k*_1_ rises, its emitted sound waves partially cancel each other out, reducing the rise. Figure 5b illustrates that the connection strength of the inner and outer shells will increase as the height of the inner shell *h*_1_ augments. Therefore, when the height of the inner shell *h*_1_ enlarges, the maximum transmitting voltage response of the transducer will gradually intensify. As shown in Figure 5c, the increment in the thickness of the inner shell *d*_1_ will raise the amplitude intensity of the outer shell driven by the active material, so that the transducer can generate more powerful sound waves and intensify the maximum transmitting voltage response of the transducer. Figure 5d demonstrates that the increase in the short axis/long axis ratio *Z*_1_ of the inner housing does not significantly change the structural stiffness of the transducer, so the maximum emission voltage does not significantly change.

Figure 6 shows the influences of various structural parameters of the inner shell on the conductance.

In Figure 6a–c, the matching between the mechanical shell and the active material of the transducer is improved to some extent with the increments in the height of the inner shell pad *k*_1_, the height of the inner shell *h*_1_ and the thickness of the inner shell *d*_1_. Therefore, when the height of the inner shell pad *k*_1_, the height of the inner shell *h*_1_, and the thickness of the inner shell d_1_ are increased, the conductivity value of the transducer experiences an upward trend. As shown in Figure 6d, as the short axis/long axis ratio *Z*_1_ of the inner housing is increased, the volume proportion of the active material will decline, and the overall impedance mode value of the transducer will increase, resulting in a lower conductance of the transducer.

Table 1 shows the rate of influence of each structural parameter of the inner shell on the conductance value of the transducer in water: the inner shell thickness *d*_1_ exerts the greatest influence on the maximum transmitting voltage response and conductance.

The formula for *f* (rate of change) can be expressed using Equation (1) where *x* is the frequency, and *y* is the maximum emission voltage or conductance:(1)$$f=\left|\frac{\Delta y_{1}}{\Delta x_{1}}\right|+\left|\frac{\Delta y_{2}}{\Delta x_{2}}\right|+\left|\frac{\Delta y_{3}}{\Delta x_{3}}\right|+\left|\frac{\Delta y_{4}}{\Delta x_{4}}\right|$$

#### 3.1.2. The Influence of Water on the Structural Parameters of the External Shell

Figure 7 shows the influences of various structural parameters of the outer shell on the maximum transmitting voltage response of the transducer.

In Figure 7a,b, because the height of the outer shell pad *k*_2_ and the height of the outer shell *h*_2_ exert little influence, the variation interval is small. As shown in Figure 7c, the maximum transmitting voltage response of the transducer intensifies, then decreases, with the increment in the outer shell thickness *d*_2_. When the outer shell thickness *d*_2_ is equal to 8 mm, the maximum transmitting voltage response of the transducer is minimized. In Figure 7d, the maximum transmitting voltage response of the transducer intensifies first and then reduces with the rising short axis/long axis ratio *Z*_2_ of the outer shell. When the short axis/long axis ratio Z_2_ of the outer shell is 0.55, the maximum transmitting voltage response of the transducer reaches a peak. When the short axis/long axis ratio *Z*_2_ of the outer shell increases, the structural stiffness of the transducer increases. It reaches the maximum when the short axis/long- axis ratio *Z*_2_ of the outer shell is 0.55, and the amplification effect of the elliptical outer shell is optimized; therefore, the maximum transmitting voltage response is maximized at this time.

Figure 8 shows the influences of various structural parameters of the outer shell on the conductance of the transducer.

As shown in Figure 8a, the conductance of the transducer descends, then ascends with the increasing height *k*_2_ of the outer shell pad, which is similar to the trend in the maximum emission voltage response. In Figure 8b, the conductance of the transducer reaches its maximum when the outer shell height *h*_2_ is 270 mm, and then begins to reduce. In the structural parameter adjustment range of the outer shell height *h*_2_, the mechanical impedance of the transducer decreases, then grows with the increment in the outer shell height *h*_2_, and the conductance of the transducer ramps up first and then declines. In Figure 8c, the conductance of the transducer rises, then declines and finally stabilizes with the increase in the thickness of the outer shell *d*_2_. This is because the modal impedance of the transducer is significantly affected by the thickness of the outer shell *d*_2_, but when it reaches a certain value, the impedance of the transducer tends to be stable. In Figure 8d, when the short axis/long axis ratio *Z*_2_ of the outer shell is equal to 0.55, the conductivity of the transducer is maximized.

Table 2 shows the rate of change in the influences of the structural parameters of the outer shell on the conductance of the transducer: the outer shell thickness *d*_2_ exerts the greatest influence on the maximum emission voltage response, and the outer shell short axis/long axis ratio *Z*_2_ exerts the largest influence on the conductance.

#### 3.1.3. Structural Parameters of Piezoelectric Ceramic Pieces in Water

Figure 9 shows the influences of various structural parameters of the piezoelectric ceramic sheet on the transmitting voltage response of the transducer.

As illustrated in Figure 9a,b, the maximum transmitting voltage response of the transducer increases slightly with the increase in the height of the piezoelectric ceramic plate *h*_p_ and the length of the piezoelectric ceramic plate *l*_p_, but the increment is not large. In Figure 9c, the maximum transmitting voltage response of the transducer decreases as the thickness of the piezoelectric ceramic plate *d*_p_ enlarges (because the larger the thickness *d*_p_ of the piezoelectric ceramic plate, the smaller the number of piezoelectric ceramic plates, the larger the capacitive reactance of the piezoelectric ceramic plate, and the smaller the amount of charge passing through the piezoelectric ceramic plate, the ability of the transducer to radiate the sound wave is reduced). Therefore, when the thickness of the piezoelectric ceramic plate *d*_p_ increases, the maximum transmitting voltage response of the transducer will drop significantly.

Figure 10 shows the influences of various structural parameters of the piezoelectric ceramic sheet on the conductance value of the transducer.

Figure 10a shows that the conductivity of the transducer remains practically unchanged with the increase in the height *h*_p_ of the piezoelectric ceramic sheet. In Figure 10b, the conductance value of the transducer grows slightly with the increase in the length *l*_p_ of the piezoelectric ceramic sheet. In Figure 10c, the conductance value of the transducer drops significantly with the increase in the thickness *d*_p_ of the piezoelectric ceramic sheet, because, when the thickness of the piezoelectric ceramic plate is larger than *d*_p_, the number of piezoelectric ceramic plates is smaller, the capacitive reactance of the piezoelectric ceramic plate is larger, the amount of charge passing through the piezoelectric ceramic plate is smaller, and the conductivity of the transducer decreases significantly.

Table 3 shows the influence of each structural parameter of the piezoelectric ceramic sheet on the conductance of the transducer: the thickness of the piezoelectric ceramic sheet *d*_p_ exerts the greatest influence on the transmitting voltage response and conductance.

The rate of change of the influence of each structural parameter of the transducer in water on the maximum transmitting voltage response and conductance value is summarized in Table 4.

Figure 11 shows the degree of influence of the transducer’s structural parameters on its maximum transmitting voltage response and conductance value in water. It can be seen that the maximum transmitting voltage of the transducer exerts a significant influence on the thickness *d*_p_ of the electroceramic plate under pressure, followed by the thickness *d*_1_ of the inner shell and the height *k*_1_ of the inner shell pad. The maximum transmitting voltage response is negatively correlated with the thickness of the piezoelectric ceramic plate *d*_p_, while it is positively correlated with the thickness of the inner shell *d*_1_ and the height of the inner shell pad *k*_1_. The conductivity of the transducer is significantly affected by the thickness of the inner shell *d*_1_ and the thickness of the piezoelectric ceramic plate *d*_p_, and is positively correlated with the thickness of the inner shell *d*_1_ and negatively correlated with the thickness of the piezoelectric ceramic plate *d*_p_.

### 3.2. Final Selection of Virtual Prototype Parameters and Performance

The variation in each structural parameter will lead to the maximum transmission response, available bandwidth, and defects in the intermediate concave valley. The main structural parameters of the final transducer prototype are listed in Table 5.

Finite element analysis was performed on the virtual optimized transducer prototype. Figure 12 shows the optimized acoustic performance parameters of the double-shell type-IV bending tension transducer. It can be seen that when the resonant frequency is 1.09 kHz, the conductance value of the transducer in the air is the largest, and the maximum conductance is 4.6 mS. When the resonant frequency is 740 Hz, the maximum conductance of the transducer in water is 0.66 mS. At the same time, when the transducer reaches resonance in water, the transmitter voltage response of the transducer is the largest, with a maximum value of 130 dB.

### 3.3. Static Analysis of Transducers

#### 3.3.1. Preload Analysis

The compression resistance of the piezoelectric ceramic plate is better than the expansion resistance, so the displacement amplification effect of the elliptical inner shell is used to extrude the piezoelectric ceramic plate to be slightly longer than the long axis of the inner shell to complete the application of pretension force.

In the finite element software, the displacement and stress of the inner shell after applying the preload were analyzed. A displacement of 0.5 mm is applied to the short axis of the structural model of the quarter inner shell. Figure 13 shows the change in the displacement and stress after the preload was applied to the inner shell.

Figure 13a displays a total displacement cloud map after the displacement of 0.5 mm is applied to the short axis of the inner shell. It can be seen that the maximum displacement of the inner shell at this time is at the short-axis end of the inner shell, and the minimum displacement is at the connection between the pad block of the inner shell and the curved shell. Figure 13b shows a displacement cloud diagram of the inner shell in the direction of the long axis. In the figure, the displacement at the end of the long axis of the inner shell is the smallest, and the long axis of the inner shell extends by 0.143 mm. Due to the axial symmetry of the analysis model of the inner shell, the long axis of the inner shell extends by 0.286 mm. Figure 13c shows a total stress cloud map after the displacement is applied. The maximum stress of the inner shell appears at the joint of the inner shell pad and the curved shell after the preload is applied, and the maximum stress is 149 MPa. The material used for the inner shell was 7075 duralumin, and its stress limit is generally above 300 MPa, which is far greater than the maximum stress on the inner shell. Therefore, when applying the preload, the force applied to the inner shell will not damage the mechanical structure.

The density of the piezoelectric ceramic sheet is much higher than that of duralumin, so the displacement of the inner shell to the piezoelectric ceramic sheet pile can be ignored after applying the preload. A displacement of 0.143 mm is applied to the pad at the long axis of the inner shell to simulate the displacement and stress changes in the inner shell after embedding the piezoelectric ceramic sheet pile. Figure 14 shows a displacement and stress cloud map of the inner shell after the piezoelectric ceramic sheet pile is embedded: after applying a displacement of 0.143 mm to the cushion block at the end of the long axis of the inner shell, the displacement diminishes at the short-axis end of the inner shell is 0.37 mm. Due to the axial symmetry of the inner shell, the short axis of the inner shell is reduced by 0.74 mm in total, and this displacement should be considered when assembling the inner shell and the outer shell. In Figure 14b, the maximum stress of the inner shell is 146 MPa, which is less than the stress limit of the inner shell and conforms to the application standard.

#### 3.3.2. Hydrostatic Pressure Analysis

Finite element analysis was carried out on the hydrostatic pressure of the outer shell, and the working depth was set to 4 m underwater and the hydrostatic pressure to 40 kPa. Figure 15 shows the hydrostatic pressure stress cloud diagram of the outer shell. It can be seen that the point of maximum stress appears at the joint of the outer shell pad and the curved shell, and the maximum stress is about 8.3 MPa, which is in line with the stress limit of the material.

## 4. Preparation

### 4.1. Preparation of the Transducer

According to the optimized structural parameters, the prototype of the double-shell type IV bending tension transducer was made. Because the transducer prototype needs to be submerged to test its acoustic performance after preparation, it is also necessary to design a sealing device for the transducer. In the present work, the transducer was sealed with epoxy resin, sealing the cover plate and fixing bolt. The specific parts of the transducer included the inner shell, the outer shell, the piezoelectric ceramic sheet, the transition block, the conductive copper sheet, the cable, the epoxy resin, the shell cover plate, and the fixing bolt.

#### 4.1.1. 3-D Printing Sand Mold

Figure 16 shows 3-d models of inner housing, outer housing, and upper and lower housing covers built using Solidworks. In the upper and lower shell cover plate design, the inner card groove structure was designed on the side of the shell cover plate and the outer shell assembly to ensure tightness.

The 3-d model (Solidworks 2022) was imported into the 3-d printer slicing software (Modellight) and printing parameters such as support, closed surface, filling type, layer height, and filling rate were set. Since the inner and outer shells were columnar structures, they were placed along the height direction of the shell to reduce the settings of the supports during slicing. The sealing surface was sealed along the 45° oblique line to improve its sealing performance. The filling utilized a grid structure, improving the structural stiffness of the printed model; the height of each layer was 0.2 mm; the filling rate was 10%. Figure 17 shows the inner shell, outer shell, and upper and lower shell cover plate models after slicing.

A 3-d printer was adopted to finish the printing; then, the 3-d printing polishing liquid was used to polish the model, and a casting mold was obtained after cleaning (Figure 18).

#### 4.1.2. Sand Casting

After placing the polished, cleaned mold into the sand box, the parting sand was sprinkled, the box was filled with casting red sand, and the casting sand was collected after stripping. Figure 19 shows the sand mold of the upper shell cover plate and the inner shell after demolding.

After the casting was finished, the castings were cut and polished to obtain the final inner shell, outer shell, upper shell cover plate, and lower shell cover plate parts.

#### 4.1.3. Assembly of Transducer Prototype

The post-treated inner shell, outer shell, upper shell cover plate, and lower shell cover plate were assembled with piezoelectric ceramic sheet, conductive copper sheet, cable, transition block, fixing bolt, and other parts. The assembly was conducted in an inside-out manner. Figure 20 shows the assembly diagram of the double-shell type-IV bending tension transducer.

The inner and outer shells were assembled using transition steel blocks, the upper and lower shell covers were connected to the inner and outer shells by four fixing bolts, and the bayonet and cable joints were filled with epoxy resin. The final assembly is shown in Figure 21.

## 5. Experiment

The double-shell type-IV bending tension transducer was submerged in water for testing, including admittance frequency response testing and transmitting voltage response measurement. The test was carried out in an anechoic pool measuring 20 × 10 × 10 m. The instruments used in the test included a WB6500B impedance analyzer, an SSG5060X signal generator, an ATA-L8 power amplifier, an HTD-21 standard hydrophone, a ZT5863K charge amplifier, and an NFE61PT472C1H9L filter. Figure 22 shows the underwater test system.

Figure 23 shows the curve of *G* and *B* components of the admittance of the transducer in water.

It can be seen that the resonant frequency of the transducer in water is 750 Hz, the conductance is 0.41 mS, the resonant frequency predicted by finite element analysis is 740 Hz, and the maximum conductance is 0.66 mS. The resonant frequency measured in water is in good agreement with the simulated results, but the conductance is slightly diminished. The analysis shows that the purity of the inner and outer shells made by sand casting is low, and other impurities are mixed into the aluminum and water due to the longer operation time during casting. As a result, the degree of matching between the shell and the piezoelectric ceramic reduces, and the conductance decreases.

## 6. Comparison

By comparing the simulation data of the transmitting voltage response with the test data (Figure 24), it can be seen that the tested maximum transmitting voltage response is 129.25 dB, while the simulated maximum transmitting voltage response is 130 dB. The emission voltage curves from the simulation and test are, in general, consistent; however, because the upper and lower shell cover plates limit the radiation displacement of the shell to a certain extent, the mechanical connection at the connection of the inner and outer shells causes a certain damping, resulting in a reduction of the radiation capacity of the transducer, and the maximum emission voltage is decreased by about 1 dB.

This result has a smaller maximum transmitting voltage response but a wider working bandwidth than the free-end cap four-beam concave broadband bending tensioner [34].

## 7. Conclusions

The current work encompasses three aspects: the proposal of a nested double-shell structure, the optimization of transducer structural parameters using finite element software, and the preparation of transducer prototype and underwater testing. The key conclusions are summarized as follows:(1)A nested double-shell structure was proposed. In view of the disadvantages of traditional low-frequency underwater acoustic transducers, such as their large structure size and difficulty in processing complex shells, a type-IV curved underwater acoustic transducer with a double-shell structure was proposed. The resonant frequency of the transducer can be reduced without increasing the structural size of the transducer;(2)The finite element software was utilized to optimize the structural parameters of the transducer. The modal analysis and harmonious response analysis of a structural model of one-eighth the size of the transducer were conducted using finite element software ANSYS (Ansys2020R2), and three acoustic properties of the transducer were obtained: resonant frequency, emission voltage response, and conductance. The influences of the structural parameters of the inner shell, outer shell, and piezoelectric ceramic plate on the acoustic performance were analyzed. The results indicate that the short axis/long axis ratio of the outer shell is proportional to the resonant frequency of the transducer, and the thickness of the outer shell is inversely proportional to the resonant frequency of the transducer. The thickness of the piezoelectric ceramic sheet is inversely proportional to the conductance of the transducer and the maximum emission voltage response. According to the observed trend, the optimized structural size of the transducer was determined, and the acoustic performance parameters of the transducer virtual prototype were obtained. The resonant frequency of the transducer virtual prototype was 740 Hz, the maximum conductivity was 0.66 mS, and the maximum transmitting voltage response was 130 dB;(3)After manufacturing the transducer mold by using the FDM printer, the transducer prototype was prepared using a sand-casting process, before being assembled and tested. The assembled transducer prototype weighed 25.6 kg; the maximum linear size was 250 mm; the resonant frequency in water was 750 Hz; the transmitting voltage response was 129.25 dB; and the conductivity was 0.41 mS. The bandwidth was 60 Hz.(4)The double-shell structure of the double-shell type-IV bending tension transducer can realize the secondary amplification of the volume displacement of the transducer during operation, which helps to lower the resonant frequency of the transducer and realize the low-frequency emission of the transducer. The combination of 3-d printing technology and sand-casting technology can accelerate the manufacturing of complex shell molds and reduce the production cycle time and production cost of transducers.

## Figures and Tables

**Figure 1 sensors-24-04746-f001:**
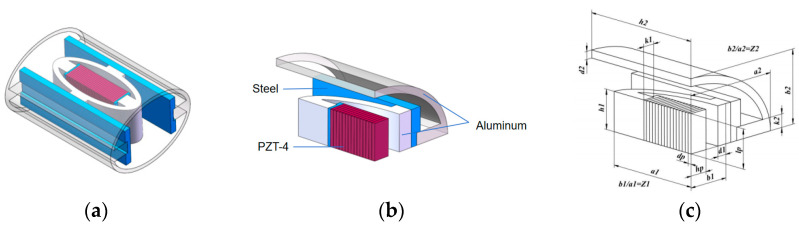
Transducer model. (**a**) Double-shell Type IV bending tension transducer; (**b**) ⅛-scale model; (**c**) schematic diagram of structural parameters.

**Figure 2 sensors-24-04746-f002:**
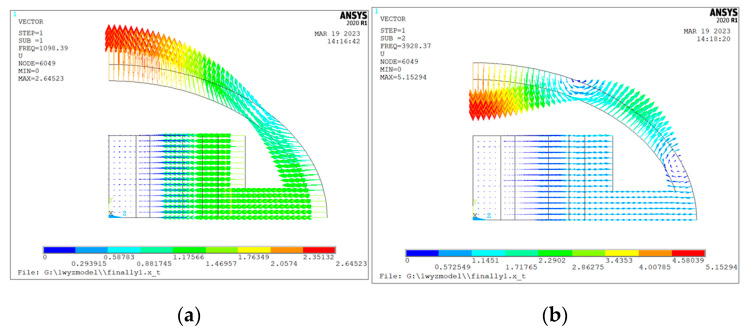
Modal analysis of ⅛-scale transducer model. (**a**) First-order mode; (**b**) second-order mode; (**c**) third-order mode; (**d**) fourth-order mode.

**Figure 3 sensors-24-04746-f003:**
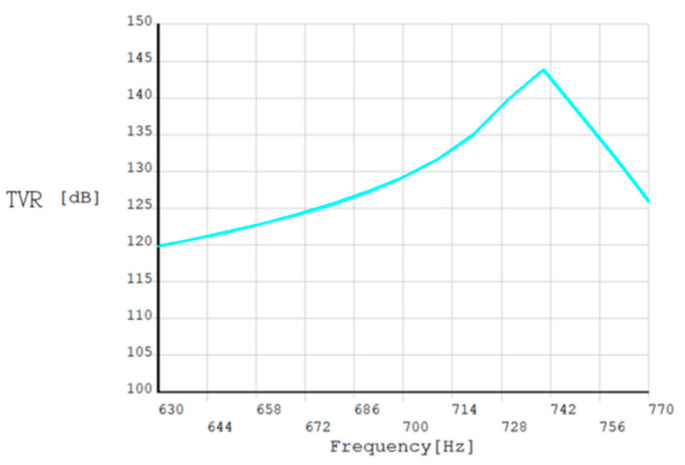
Transmitting voltage response curve of transducers.

**Figure 4 sensors-24-04746-f004:**
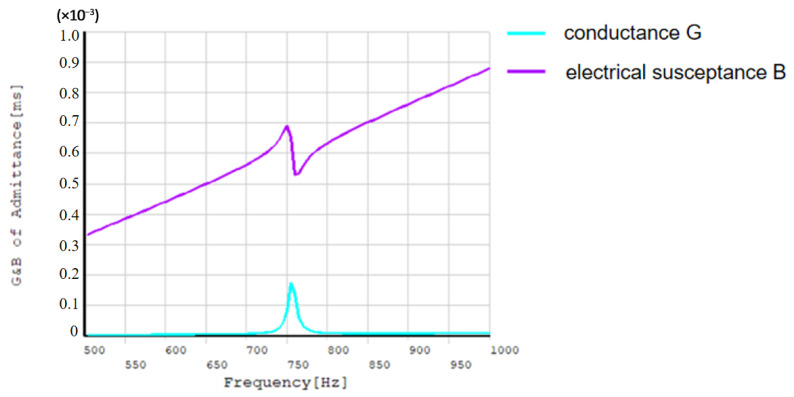
The admittance *G* and *B* component curve of the transducer in water.

**Figure 5 sensors-24-04746-f005:**
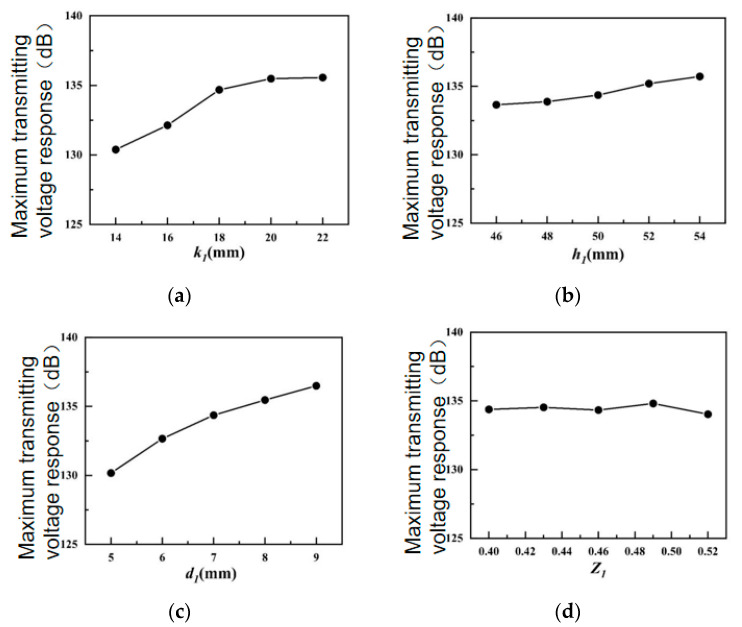
Influences of structural parameters of inner shell in water on the maximum TVR. (**a**) Inner housing pad height; (**b**) inner housing height; (**c**) inner shell thickness; (**d**) inner shell short axis/long axis ratio.

**Figure 6 sensors-24-04746-f006:**
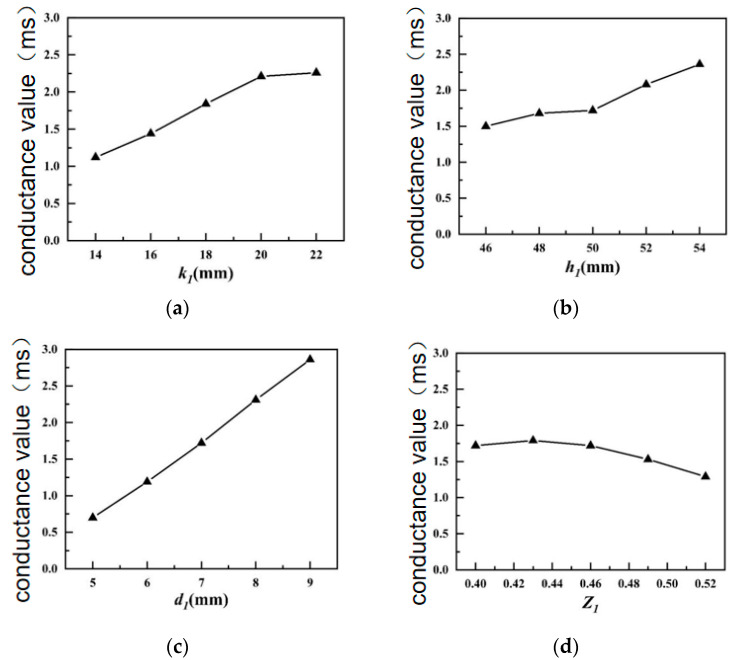
Influences of structural parameters of the inner shell in water on the conductivity. (**a**) Inner housing pad height; (**b**) Inner housing height; (**c**) inner shell thickness; (**d**) inner shell short axis/long axis ratio.

**Figure 7 sensors-24-04746-f007:**
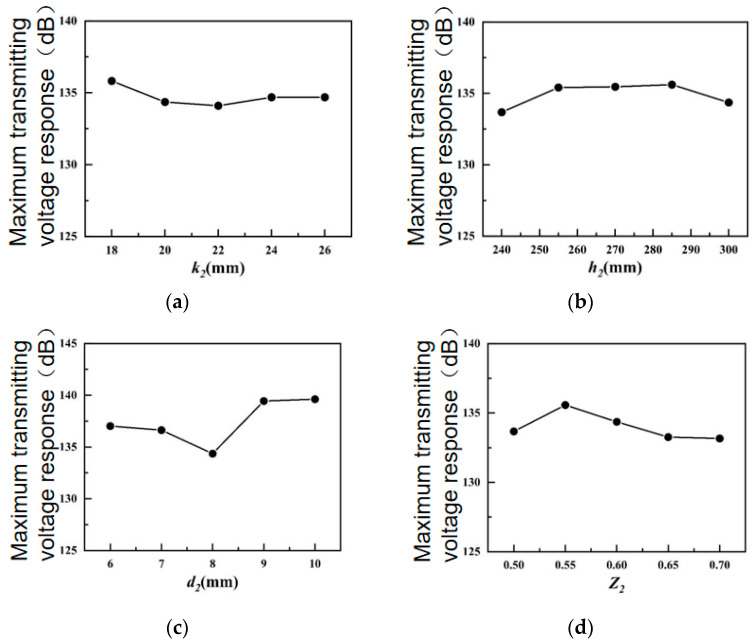
Influences of structural parameters of underwater shell on the maximum TVR. (**a**) Height of outer housing pad; (**b**) height of outer housing; (**c**) outer shell thickness; (**d**) outer shell short axis/long axis ratio.

**Figure 8 sensors-24-04746-f008:**
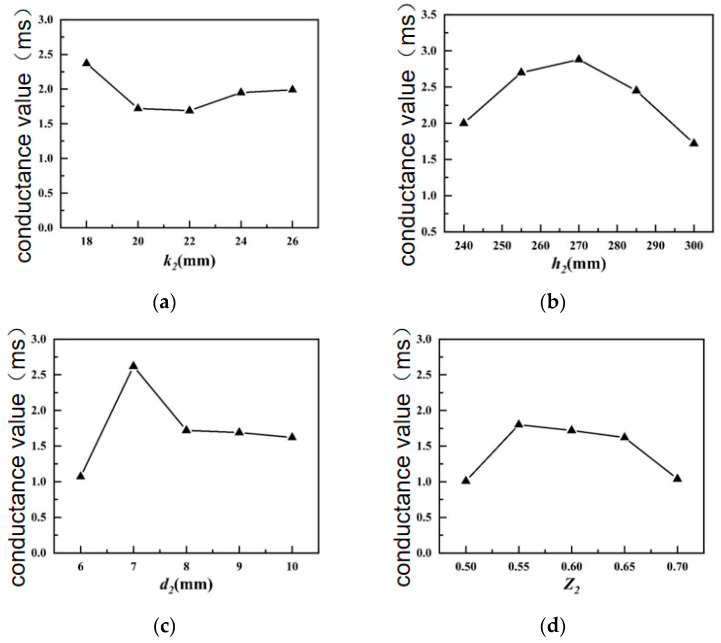
Influences of structural parameters of underwater shell on conductivity value. (**a**) Height of outer housing pad; (**b**) height of outer housing; (**c**) outer shell thickness; (**d**) outer shell short axis/long axis ratio.

**Figure 9 sensors-24-04746-f009:**
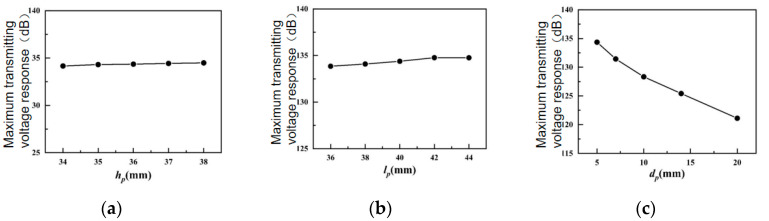
Influences of structural parameters of piezoelectric ceramics in water on the maximum TVR. (**a**) Piezoelectric ceramic sheet height; (**b**) length of piezoelectric ceramic sheet; (**c**) thickness of piezoelectric ceramic sheet.

**Figure 10 sensors-24-04746-f010:**
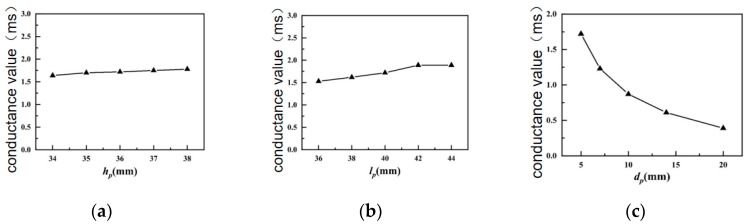
Influences of structural parameters of piezoelectric ceramic sheets in water on the conductivity value. (**a**) Piezoelectric ceramic sheet height; (**b**) Length of piezoelectric ceramic sheet; (**c**) thickness of piezoelectric ceramic sheet.

**Figure 11 sensors-24-04746-f011:**
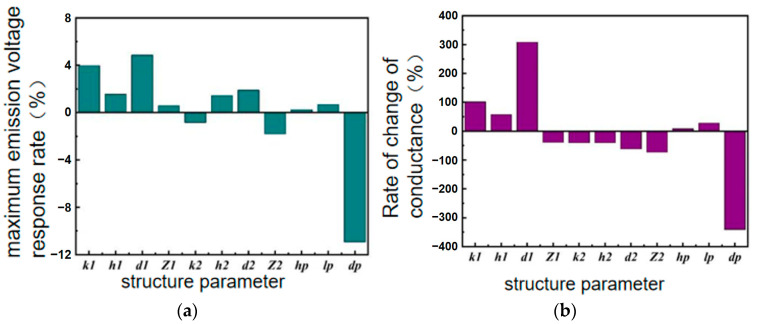
Influences of structural parameters of underwater transducers on acoustic performance. (**a**) Change in maximum emission voltage response; (**b**) change in conductance.

**Figure 12 sensors-24-04746-f012:**
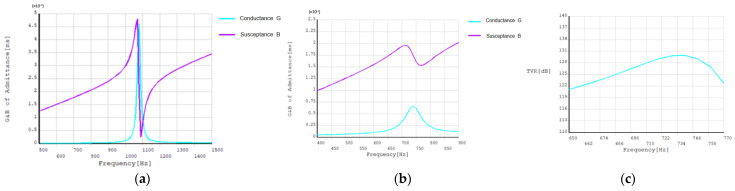
Optimized acoustic performance. (**a**) The optimized admittance *G* and *B* components in the air; (**b**) The optimized admittance value *g* and *b* components in water; (**c**) the optimized emission voltage response.

**Figure 13 sensors-24-04746-f013:**
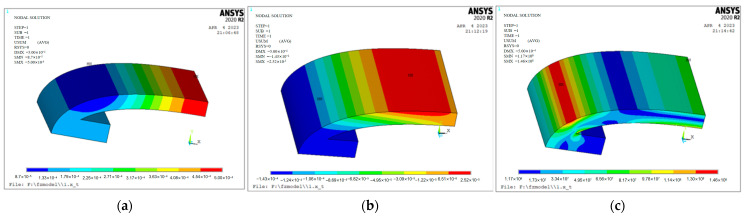
Displacement and stress cloud diagram of the inner shell. (**a**) Total displacement cloud map; (**b**) long-axis displacement nephogram; (**c**) total stress nephogram.

**Figure 14 sensors-24-04746-f014:**
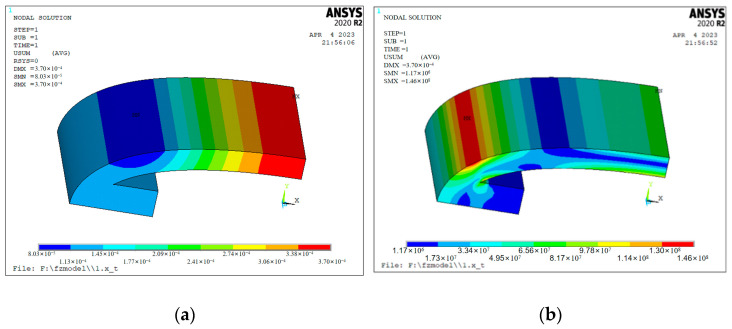
Displacement and stress cloud diagram of the long-axis end of the inner shell. (**a**) Displacement cloud image in the long-axis direction; (**b**) stress cloud image in the long-axis direction.

**Figure 15 sensors-24-04746-f015:**
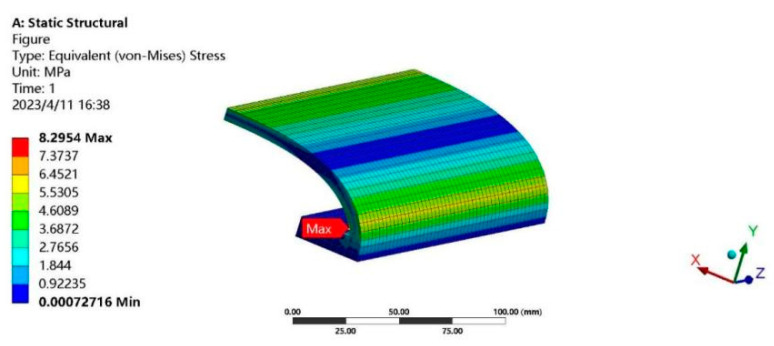
Hydrostatic pressure cloud diagram of shell body.

**Figure 16 sensors-24-04746-f016:**
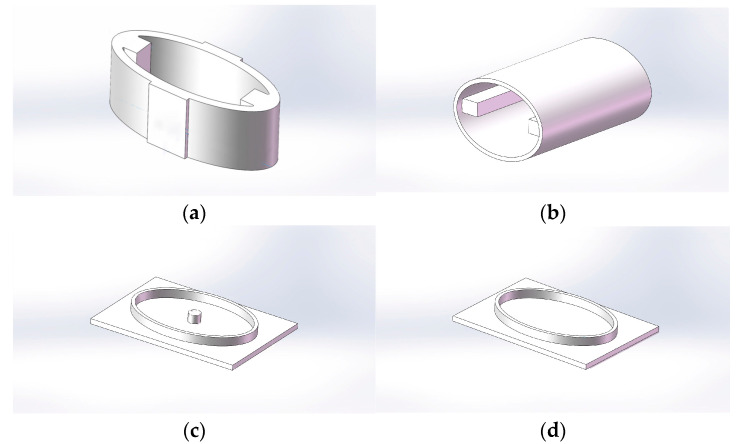
Three-dimensional molds. (**a**) Inner shell; (**b**) outer shell; (**c**) upper housing cover plate; (**d**) lower housing cover plate.

**Figure 17 sensors-24-04746-f017:**
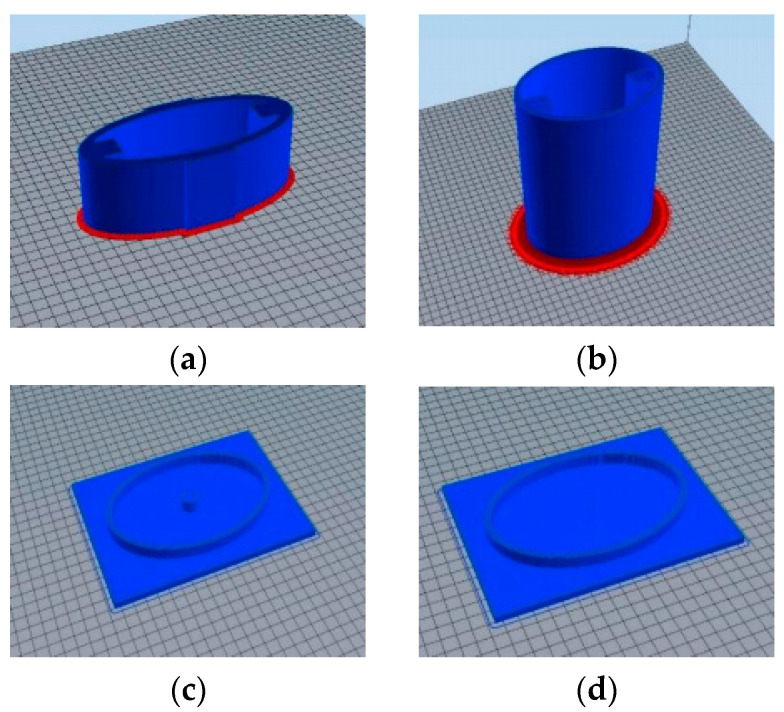
Three-dimensional slice models. (**a**) Inner shell; (**b**) outer shell; (**c**) upper housing cover plate; (**d**) lower housing cover plate.

**Figure 18 sensors-24-04746-f018:**
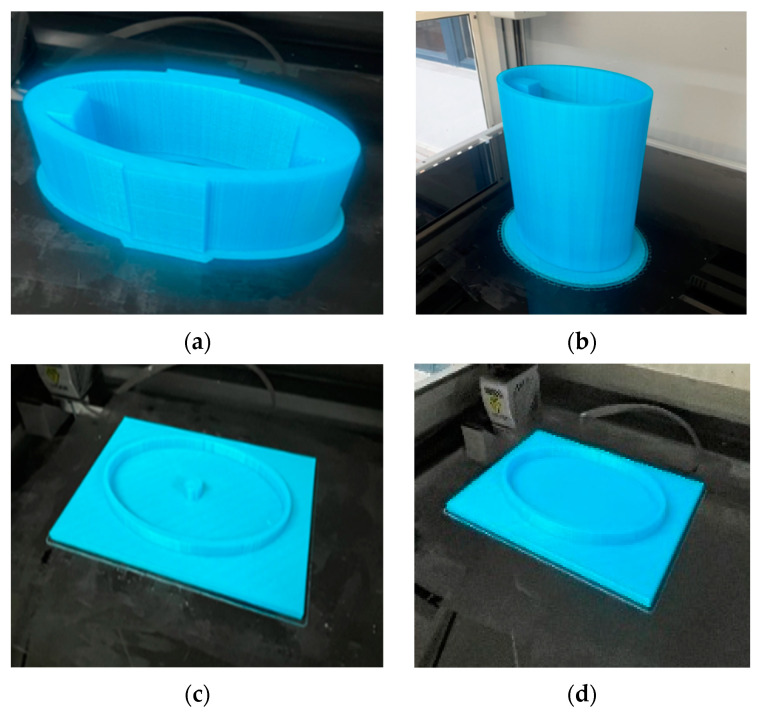
Three-dimensionally printed molds. (**a**) Inner shell; (**b**) outer shell; (**c**) upper housing cover plate; (**d**) lower housing cover plate.

**Figure 19 sensors-24-04746-f019:**
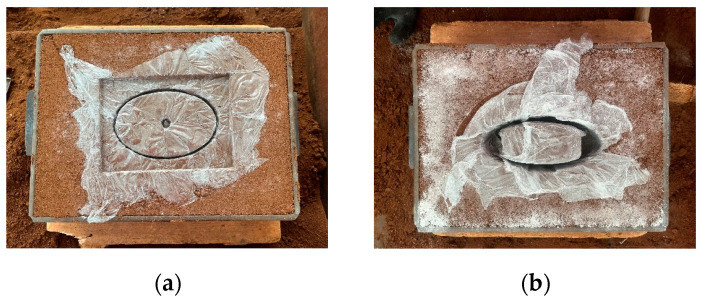
Sand mold after demolding. (**a**) Upper housing cover plate; (**b**) inner housing.

**Figure 20 sensors-24-04746-f020:**
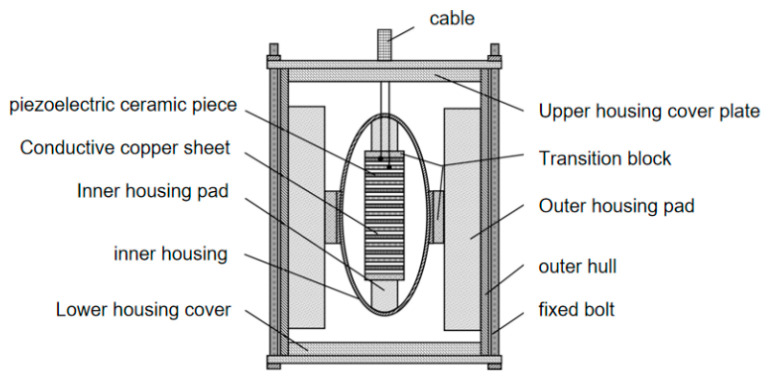
General assembly drawing of dual-shell class-IV flextensional transducers.

**Figure 21 sensors-24-04746-f021:**
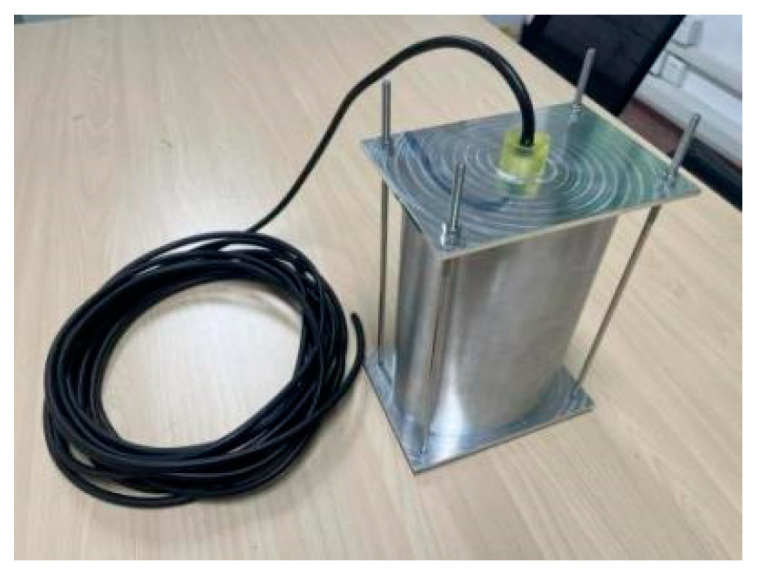
Prototype of dual-shell class-IV flextensional transducers.

**Figure 22 sensors-24-04746-f022:**
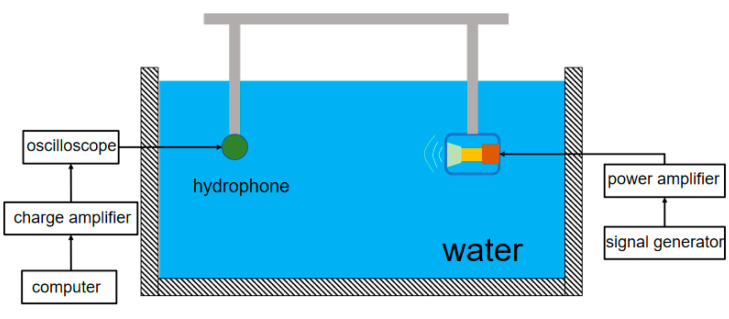
Testing system.

**Figure 23 sensors-24-04746-f023:**
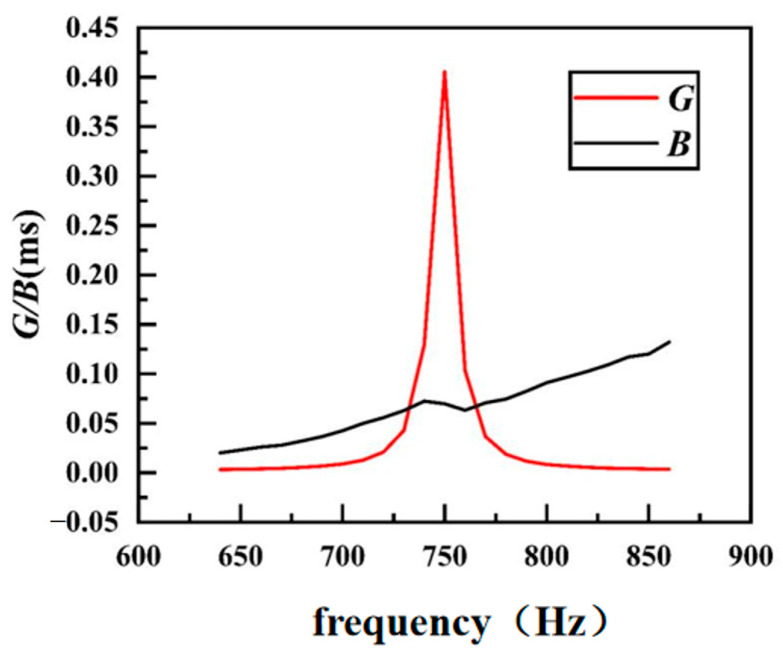
Admittance values of *G* and *B* components in water.

**Figure 24 sensors-24-04746-f024:**
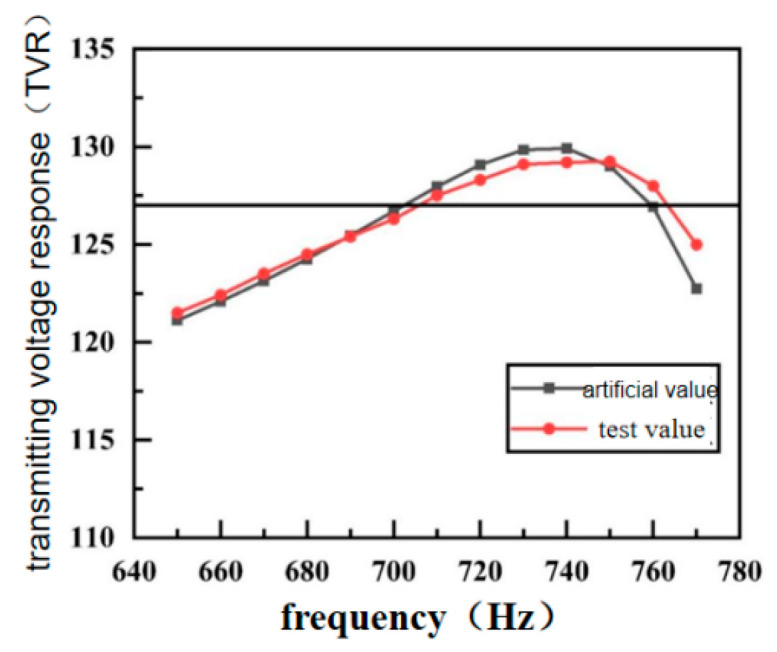
Test and simulation values of transmitting voltage response curve.

**Table 1 sensors-24-04746-t001:** Rate of change in the influence of structural parameters of the inner shell in water.

**Structural Parameter**	*k* _1_	*h* _1_	*d* _1_	*Z* _1_
**Maximum transmitting voltage response (%)**	3.97	1.55	4.86	0.58
**Conductance (%)**	101.79	57.33	308.27	38.76

**Table 2 sensors-24-04746-t002:** Rate of change in the influences of structural parameters of the underwater shell.

**S** **tructur** **al** **Parameter**	*k* _2_	*h* _2_	*d* _2_	*Z* _2_
**T** **ransmitting voltage response (%)**	0.84	1.44	1.90	1.81
**C** **onductance (%)**	40.24	40.12	61.73	73.08

**Table 3 sensors-24-04746-t003:** Rate of change in the influences of structural parameters on piezoelectric ceramic sheets in water.

**Structural Parameter**	*h* _p_	*l* _w_	*d* _p_
**Transmitting voltage response (%)**	0.25	0.68	10.93
**Conductance (%)**	8.54	27.69	341.03

**Table 4 sensors-24-04746-t004:** Rate of change in the influences of structural parameters of underwater transducers.

Structural Parameter	Transmitting Voltage Response (%)	Conductance (%)
*k* _1_	3.97	101.79
*h* _1_	1.55	57.33
*d* _1_	4.86	308.27
*Z* _1_	0.58	38.76
*k* _2_	0.84	40.24
*h* _2_	1.44	40.12
*d* _2_	1.9	61.73
*Z* _2_	1.81	73.08
*h* _p_	0.25	8.54
*l* _p_	0.68	27.69
*d* _p_	10.93	341.03

**Table 5 sensors-24-04746-t005:** Optimized structural parameters.

Structural parameter	*k* _1_	*h* _1_	*d* _1_	*Z* _1_	*k* _2_	*h* _2_	*d* _2_	*Z* _2_	*h* _p_	*l* _p_	*d* _p_
Quantitative value (mm)	20	60	8	45%	20	200	6	70%	30	60	3

## Data Availability

Data are contained within the article.

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
