# Peer review of "Structural Optimization and Performance of a Low-Frequency Double-Shell Type-IV Flexural Hydroacoustic Transducer"

_sensors, 2024, doi:10.3390/s24144746_

Round 1

Reviewer 1 Report

Comments and Suggestions for Authors

This paper studied the structure optimization and performance of low-frequency double-shell Type IV flexural hydroacoustic transducer. This paper is well organized and is solid with detailed description. I therefore recommend its publication in Sensors. Here are some issues to be clarified to improve the study.

 1- The introduction part is short, it would benefit from a more detailed review of existing literature to better contextualize the novelty of your approach.

2- The discussion could be expanded to compare the findings with previous studies more explicitly. For example, give a comparison Table between this work and other works.

 3-Could the author provide more description about 3D Printing Sand Mold in part 3.1.1.? Since it is an important innovation point of this manuscript.

4. The quality of figures need to be improved. The labeling in figures needs to be uniform. In Figure12, is labeling G&B written in Chinese?

5- In Line 457 “the mechanical Q value is 12.5”. This description does not come up in the discussion section.

Comments on the Quality of English Language

Minor editing of English language required

Author Response

I have improved the quality of english language.

1.The introduction has been added.

'The nested double shell structure is proposed, aiming at the disadvantages of traditional low frequency underwater acoustic transducer, such as large structure size and difficult to process complex shell, a type IV curved underwater acoustic transducer with double shell structure is proposed. The resonant frequency of the transducer can be reduced without increasing the structure size of the transducer. '

2.A comparison is added at the end of the article, although the size of the two transducers is different.

3.Added some work and data before 3D printing.

4.The label in Figure 12 has been modified.

5. In Line 457 “the mechanical Q value is 12.5”. This description has been deleted.

Reviewer 2 Report

Comments and Suggestions for Authors

  In this paper, the author proposed a kind of Type IV flexural hydroacoustic transducer. By using 3D printing technic, the transducer is easy to be built. The organization of the manuscript is appropriate and the results are good. I think this paper is appropriate to publish in ‘Sensors’. And also, there are some typos, grammar and errors. For example:

1.     There are two Figure 2 in this paper. The first one is not clear enough. The difference between materials should be point out in the figure. And also, the key parameters used in part 2 should be shown in this figure.

2.     I the second Figure 2, there are four kinds of vibration modes. The resonant frequencies are 1098.39Hz, 3928.37Hz, 4160.3Hz and 5576.99Hz. But in Figure 3 the underwater resonant frequency is about 740Hz. The decrease of resonant frequency should be explained.

3.     Actually, for this kind of transducer, lots of researchers like to make the first two vibrations coupled together to broaden the width of transducer.  

4.     In table 2, the authors give out a concept of ‘change rate’, but there is no explanation how to calculate it.

5.     In table 5, the optimized structural parameters have been given out, but why this kind of combination is optimized?

6.     In page 14 line 424, ‘meet job requirements’ should be deleted.

7.     In part 5, the key parameters of transducer (such as frequency, TVR, Bandwidth, weight) should be compared with some new references.

  In a word, this paper needs a major revision.

Comments on the Quality of English Language

no comments.

Author Response

I have improved the quality of english language.

1.Figure 1 and Figure 2 have been distinguished.

   The materials in each part of the figure have been marked.

   The sizes used below are shown in the diagram.

2.Add explanation:

'Because water is much denser than air, the additional mass produced is greater, so it will cause the natural frequency to decrease. '

3.Thanks for your guidance. Further research will be conducted in this direction which makes the first two vibrations coupled together to broaden the width of transducer.  

4.A formula for the rate of change was added to the article.

5.Add explanation:

'Because of the variation of each structural parameter will lead to the maximum transmission response, available bandwidth, and the defect of intermediate concave valley.'

6.In page 14 line 424, ‘meet job requirements’ has been deleted.

7.A comparison between the transducers is added. Although the size of the two transducers is slightly different, it can be found that the working bandwidth of this paper is slightly better.

Round 2

Reviewer 2 Report

Comments and Suggestions for Authors

no more comments.

Comments on the Quality of English Language

no more comments.